# Empirical Assessment of the Quality of MVC Web Applications Returned by `xGenerator`

**Gaetanino Paolone** [1], **Romolo Paesani** [1], **Martina Marinelli** [1] and **Paolino Di Felice** [2,*]

1 Gruppo SI S.c.a.r.l., 64100 Teramo, Italy; g.paolone@softwareindustriale.it (G.P.);
r.paesani@softwareindustriale.it (R.P.); m.marinelli@softwareindustriale.it (M.M.)
2 Department of Industrial and Information Engineering and Economics, University of L'Aquila,
67100 L'Aquila, Italy
* Correspondence: paolino.difelice@univaq.it; Tel.: +39-320-423-2540

**Abstract:** Many scholars have reported that the adoption of Model Driven Engineering (MDE) in the industry is still marginal. Real-life case studies, completed with convincing empirical data about the quality of the developed source code, is an effective way to persuade the industry that the adoption of MDE brings an actual added value. This paper reports about the assessment of the quality of the code outputted by `xGenerator`: a Java technology platform for the development of enterprise Web applications, which implements the MDE paradigm. Two recent papers from Aniche and his colleagues were selected to carry out the measurements. The former study is about metrics and thresholds for MVC Web applications, while the latter presents a catalog of six smells tailored to MVC Web applications. A big merit of both of these proposals is that they fix the metric thresholds by taking into account the MVC software architecture. The results of the empirical assessment, carried out on a real-life project, proved that the quality of the code is high.

**Keywords:** code metric; Model Driven Architecture; Model Driven Engineering; Model-View-Controller; smell; UML; `xGenerator`; Web application





## 1. Introduction

"Model Driven Architecture (MDA) is an approach to software design, development and implementation by the OMG. MDA provides guidelines for structuring software specifications that are expressed as models. MDA separates business and application logic from the underlying platform technology" (sentences taken from: https://www.omg.org/mda/). MDA has been promoted to solve one of the main problems faced by the software industry: coping with the complexity of software development by raising the abstraction level and introducing more automation in the process. MDA enables model-driven software development which treats models as primary development artefacts.

Ref. [1] reports about the state-of-the-art of code generation using MDA. The authors identified 50 primary studies out of 2145 related MDA articles over the period 2008–2018. From the studies published in major journals and international conferences, it comes out that until now scholars have been focused predominantly on:

- modeling languages (e.g., [2]). Here, the open debate is domain-specific modeling languages versus general-purpose modeling languages (e.g., [3,4]);
- Model-To-Model and Model-To-Text transformations (e.g., [5,6]);
- tools supporting the transformation techniques (e.g., [5,7]). In 2019, Kahani et al. [5] identified 60 tools based on the used transformation approach. Their study provides an up-to-date, in depth picture on this topic;
- the state of applying MDE in the industry and the factors considered relevant for its adoption (e.g., [8–12]).

Very recently, Bucchiarone et al. [13] reported that the adoption of MDE in the industry is still marginal. Of course "research in MDE is useless without having industrial adoption",

ref. [8] (p. 41). Real-life case studies, completed with convincing empirical data about the quality of the developed source code, are an effective way to persuade the industry that the adoption of MDE brings an actual added value. This is the core of the present paper. In fact, it reports about the assessment of the quality of the code outputted by `xGenerator`. The latter is a proprietary Java technology platform for the creation of enterprise Web applications. `xGenerator` complies with the Software Development Process described in [14]. At a high level of abstraction, such a tool acts as a black box that receives as input a UML model and returns the Java code of the Web application. The present empirical study completes the previous research by adding the part missing in that paper, namely the assessment of the quality of the generated code. Hence, this study constitutes an integral part of the previous one. The assessment of the quality of the code is based on two quite recent papers from Aniche and his colleagues [15,16]. The former study is about *metrics* and *thresholds* for MVC Web applications, while the latter presents a catalog of six smellstailored to MVC Web applications. Generally speaking, code metric analysis is the common method for assessing the quality of programs; while code smell studies propose ways of detecting violations of coding design principles together with refactoring actions to enhance the quality. The paper is structured as follows. Section 2 introduces notions and terms used throughout the paper. In particular, the section (a) provides an overview of the Software Development Process implemented by `xGenerator`, (b) recalls the basic features of such a tool and (c) surveys many papers about the assessment of the quality of object-oriented source code. A real-life case study is the focus of Section 3, while Section 4 describes the method adopted to carry out the study. Section 5 presents and discusses the results of the experiments obtained by adopting an open source tool (`Springlint`). Section 6 concludes the paper.

## 2. Background

This section recalls arguments at the core of the present paper. It starts with a light description of MDA, code generation and the Web Model-View-Controller architecture, then basic concepts from [14] are recalled and eventually pertinent studies on code metrics and code smells are surveyed.

**Model Driven Architecture** MDA provides guidelines for structuring software specifications that are expressed as models [17]. The MDA is structured in terms of three models: the Computation Independent Model (CIM), the Platform Independent Model (PIM) and the Platform Specific Model (PSM). A transformation converts models from one level of abstraction to another, usually from a more abstract to less abstract view, by adding more detail supplied by the transformation rules. Transformations can be Model-To-Model (it concerns the transition from CIM to PIM or from PIM to PSM) and Model-To-Text (it concerns the generation of the code from the PSM to a specific programming language as a target). Ref. [1] reports about the state-of-the-art of code generation using MDA research in software engineering.

**Code generation** Code generators output, from a model, the full code or a template to be completed with handwritten code [18]. The *Generation gap pattern* [19] is a technique able to keep the code and model consistent after a regeneration even when both have been modified. The strategy isolates the generated code by means of inheritance: the handwritten classes extend the generated ones. In case of regeneration, the code generator can safely overwrite the superclasses, while those manually written remain unaffected.

**The Model-View-Controller (MVC)** The MVC pattern for enterprise Web applications is given in Figure 1. The rectangles with rounded corners are part of the standard pattern [20]. Each layer is composed of classes that implement the interactions among the layers. Model implements the business logic of the application and includes the methods for connecting to the database. Controller interacts with Model in order to retrieve the needed data and generate the views. View is responsible for the way data are displayed and how users interact with it. Moreover, it supports the data gathering from the users.

Browser displays the HTML statements created by the View classes and sends requests to the Controller.

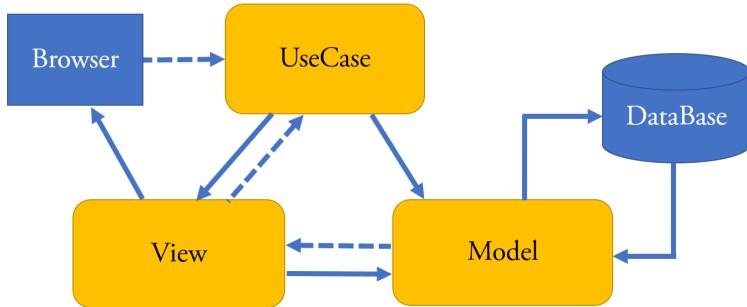

**Figure 1.** The standard Model-View-Controller (MVC) pattern of Web applications.

## 2.1. Overview of the Software Development Process

The process aims at automating the development of enterprise Web applications. It is UML-based, implements the MDA and makes use of a proprietary tool (`xGenerator`). At the CIM level, both *structural* and *behavioral* aspects of the company business are modeled, as well as the relationships between them. The transformation of models used focuses on the *Use Case* (UC) and *Class* constructs which, together, cover both the behavioral and the structural aspects. In the Software Development Process, the UC "enters the scene" at the CIM level to model the business and becomes a Java class in the code of the Web application. Figure 2 shows the different levels of abstractions of the UCs across the MDA's layers. At the CIM, the UCs are called Business UC Realization, while at the PIM they are called System UC Realization; according to the Rational Unified Process. The UC construct is also present on the third layer, where it maps the PSM to (Java) code through code generation. Figure 3 summarizes the approach proposed in [14].

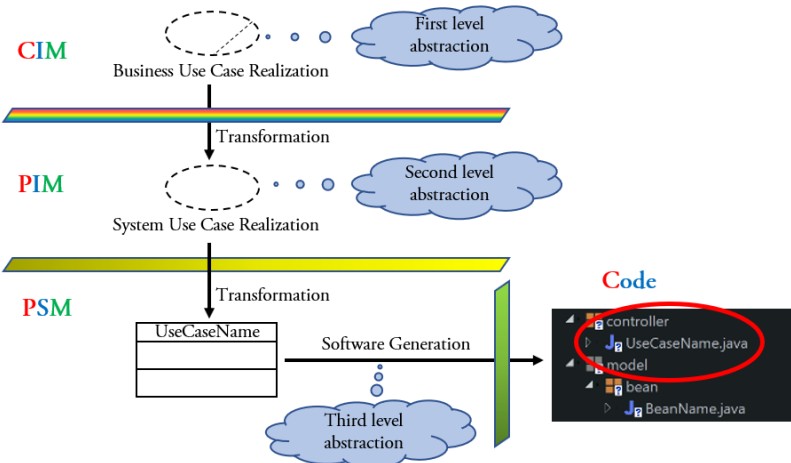

**Figure 2.** The Use Case (UC) abstraction levels across the Model Driven Architecture (MDA) layers of the Software Development Process in [14].

## 2.2. An Overview of xGenerator

This tool supports the Software Development Process proposed in [14]: analysts and designers develop the artefacts of CIM, while those of PIM and PSM, as well as the source code, are generated. The source Java code of the Web application is returned in the `Eclipse` environment. In accordance with the *Generation gap pattern* [19], the classes are structured at two different levels of abstraction: for each class of the PSM, it is generated a superclass and a subclass. The artefacts of the code model are: (a) one Java class for each class of

the PSM at each architectural layer; (b) a Java superclass for each class of the PSM at each architectural layer and a subclass that inherits from the superclass.

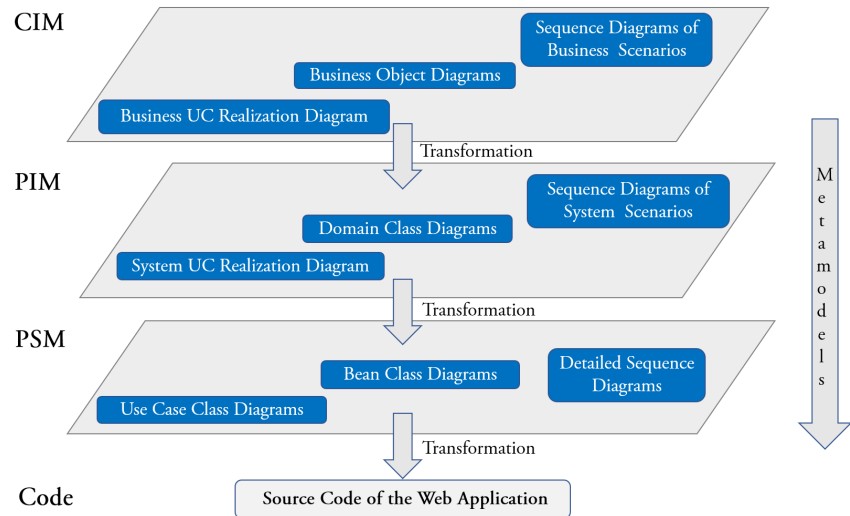

**Figure 3.** Overview of the MDA transformational approach in [14].

Each Java class returned by `xGenerator` focuses on a specific task of the MVC layer the class belongs to (*high cohesion*). The classes (returned by `xGenerator`) belonging to different layers of the MVC architecture are independent of each other (*low coupling*). Cohesion and coupling relate to the relationships that exist within a class and between classes, respectively (more in [21,22]). High cohesion and low coupling are desirable features of object-oriented software, e.g., [23]. The classes belonging to the same layer of the MVC pattern are structurally identical. This does not necessarily happen if the classes are coded by a team of programmers. Such a feature of classes of Web applications has direct impact on the code readability that, in turn, is an essential characteristic of code quality [24].

Figure 4 shows the architecture of the enterprise Web applications that can be developed with `xGenerator`. This architecture instantiates the basic MVC pattern. `Bean` and `QueryContainer` classes belong to the Model layer. The `Bean` classes are mapped to database's tables through the `Resource` component that wraps `Hibernate`. The `QueryContainer` classes define the filter criteria in the queries to be executed against the database. View is composed of three class sub-layers: `ViewBeanInfo` displays (in a panel) the information of a `Bean`, `ViewQueryInfo` displays (in a panel) the query criteria and `UseCasePanel` sends the information to the browser. Controller is composed of the layer of the `UseCases` classes; the latter implement the standard behavior of use cases and the navigation of the panels they are composed of.

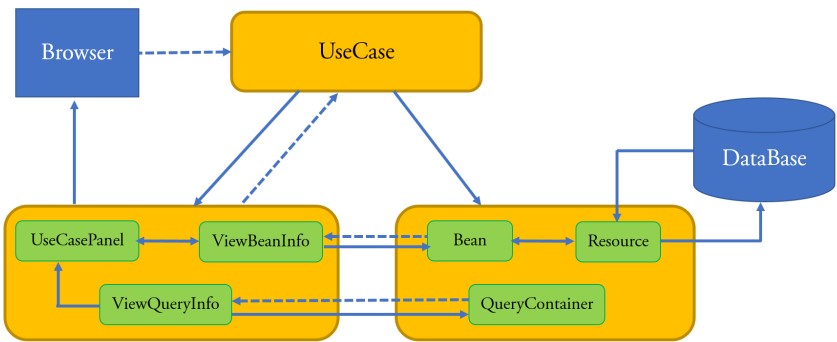

**Figure 4.** The architecture of the enterprise Web applications developed with `xGenerator`.

### 2.3. Code Metrics and Code Smells

Code metric analysis is the common method for assessing the quality of a software system. Kent Beck coined the term "code smell" in the context of identifying quality issues in object-oriented source code [25]. Code smells is one of the symptoms of low-quality code that, therefore, is claiming for improvement. Code smells do not prevent the software from working; however, they denote weaknesses in the design part of the system.

The recent study by Tahir et al. [26] has pointed out that there is a high debate between developers, across technical Stack Exchange sites, on code smells, their meaning and their impact. As an obvious consequence, the number of empirical studies on the topic is rising constantly. The smell metaphor has been adopted for several categories of software projects. Alkharabsheh et al. [27] carried out a systematic analysis of the state-of-the-art about code smell detection spanning over the period 2000–2017 (the smell metaphor was introduced in 1999 by Kent Beck). Sharma and Spinellis [28] is another up-to-date survey on studies about smell detection methods published in the period 1999–2016. Kaur [29], in turn, conducted a literature review that assessed and reported the findings of empirical studies, published till March 2018, about the impact of code smells on software quality.

The code smell detection analysis can be carried out by taking into account structural, historical and semantic properties. In the first case, the smell detection task exploits metrics in order to describe the structure of the code. A well-known suite of object-oriented metrics is given in [21].

Code smells impact understandability, maintainability, testability, complexity, performance, functionality, reusability and change-proneness of the software. Moreover, smells may increase the effort (and hence the cost) required to produce the code. Table 1, built from the data in [28], lists the five smell detection methods mentioned in such a paper together with the number of surveyed studies. From the table, we see that the metrics-based method is the most investigated by scholars.

**Table 1.** Smell detection methods and the number of surveyed studies in [28].

| Smell Detection Method | Number of References |
| --- | --- |
| Metrics-based | 19 |
| Rule/Heuristics-based | 15 |
| Machine learning-based | 6 |
| Optimization-based | 4 |
| History-based | 2 |

Metrics-based smell detection methods are relatively easy to be implemented; however, a non-trivial challenge posed by those methods is the choice of the thresholds, as pointed out by the software engineering community (see, for instance, [30,31]). On this point, Lacerda et al. [32] write the following: "There is no consensus on the standard threshold values for the detection of smells, which are the cause of the disparity in the results of different approaches." Fontana et al. [33] pointed out that the assessments produced by metrics-based smell detection tools are prone to high false-positive rates because, as already said, this category of methods depends on the metrics thresholds. In the metrics-based methods, the false-positives cannot be eliminated until the context is taken into account, and this is because one set of thresholds do not hold good necessarily in another context. According to Gil and Lalouche [34] "the code metric values, when inspected out of context, mean nothing"; in fact, they proved that metric values vary among projects.

Until a few years ago, software tools and published proposals did not take the system architecture into account. This means that all classes within an object-oriented application were treated as they were equal to each other and, hence, assessed in the same way, regardless of their specific *architectural role*. This approach is not satisfactory if applied to an MVC Web application, where Controller classes are quite different from Model classes simply because they play very different roles: the former are responsible for coordinating the flow between the View and the Model layers, while the latter implements business concepts.

Adding context to code metrics is a recent research topic. In Aniche et al. [15], the authors adapt the threshold of metrics to the "class's architectural role" with regard to the (Spring) MVC pattern (Figure 5); where the *architectural role* is defined as the particular role that a class plays in a given system architecture. Specifically, Aniche et al. focus on the server-side code, namely on the Controller and Model layers. Ref. [15] relies on the Chidamber and Kemerer metrics suite [21], as it covers different aspects of object-oriented programming, such as coupling, cohesion and complexity.

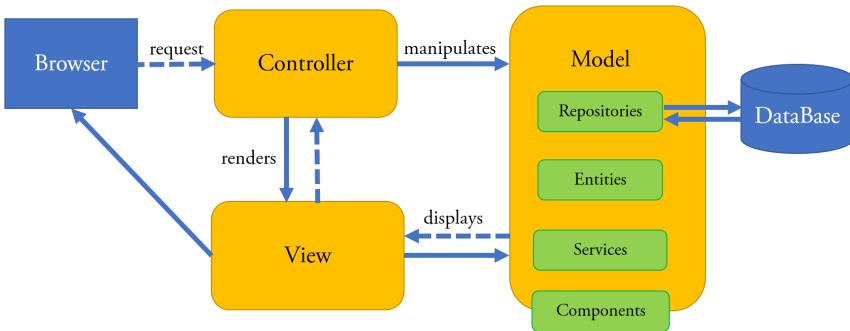

**Figure 5.** The Spring MVC architecture as depicted in [16].

Table 2 shows the class level metrics taken from [21], while Table 3 shows the thresholds found by Aniche et al. [15] for the classes of the five architectural role belonging to the Controller and Model layers. Each metric varies from 0 or 1 to infinite. The triple $[v_1, v_2, v_3]$ denotes, in order, the low, high and very high thresholds corresponding to moderate/high-/very high risk. Classes in which metric values range in the 70–80% have "moderate risk", while from 80–90% the risk is "high" and "very high" between 90–100%. A percentile is a measure used in statistics indicating the value below which a given percentage of observations in a group of observations falls.

**Table 2.** The reference class level metrics in [15].

| Metric | Acronym | Meaning |
|---|---|---|
| Coupling between Object Classes | CBO | The number of classes a class depends upon. It counts classes used from both external libraries as well as classes from the project. |
| Lack of COhesion in Methods | LCOM | The lack of cohesion in a class counts the number of intersections between methods and attributes. The higher the number, the less cohesive is the class. |
| Number Of Methods | NOM | Number of methods in a class. |
| Response For a Class | RFC | It is the count of all method invocations that happen in a class. |
| Weighted Methods per Class | WMC | Sum of McCabe's cyclomatic complexity for each method in the class. |

In a more recent paper, Aniche et al. [16] investigated the link between a set of metrics and code-smell detection. Several studies have been carried out adopting the catalog of 22 code smells defined by Martin Fowler and Kent Beck in the Refactoring book [25], and including smells that fit well in any object-oriented software system. The code smells from [25] capture general principles of good design, while they ignore the actual architecture of the software or the role played by each class inside the application. Aniche et al. [16] argued that it is possible to discover bad practices on software systems adopting a specific architecture by taking into account specific types of code smells. In concrete, they proposed a catalog of six smells tailored to MVC Web applications. Their

findings show that the adopted smells have more chances of being subject to changes and defects; moreover, the smells survive for long time in the system.

**Table 3.** The thresholds for the reference class level metrics of Table 2.

| Architectural Role | CBO | LCOM | NOM | RFC | WMC |
|---|---|---|---|---|---|
| Controller | [26,29,34] | [33,95,435] | [16,22,37] | [62,78,110] | [57,83,130] |
| Repository | [15,21,31] | [36,106,351] | [19,26,41] | [50,76,123] | [60,104,212] |
| Service | [27,34,47] | [133,271,622] | [23,32,45] | [88,123,190] | [97,146,229] |
| Entity | [16,20,25] | [440,727,1844] | [33,42,64] | [8,12,25] | [49,61,88] |
| Component | [20,25,36] | [50,123,433] | [15,22,35] | [56,81,132] | [52,79,125] |

Table 4 reports the six MVC smells of the catalog in [16], while Table 5 collects the thresholds used to detect them. To know the motivations behind the adoption of those smells and the detection strategy adopted to spot them, please refer to such a paper.

By exploiting the catalog of smells in [16], this allows to overcome the critical issue pointed out by Hozano et al. [35]: "the informal and subjective definition of certain smell types [...] may lead two or more developers to reason about each smell occurrence differently."

**Table 4.** The MVC smells in [16].

| Smell | Description |
|---|---|
| Promiscuous Controller | Offer too many actions |
| Brain Controller | Too much flow control |
| Meddling Service | The service directly query the database |
| Brain Repository | Complex logic in the repository |
| Laborious Repository Method | A method having multiple actions |
| Fat Repository | A repository managing too many entities |

**Table 5.** The thresholds used to detect the smells in Table 4.

| Metric | | Threshold |
|---|---|---|
| Promiscuous Controller | | |
| | Number of Routes (NOR) | 10 |
| | Number of Services as Dependencies (NSD) | 3 |
| Brain Controller | | |
| | Non-Framework RFC (NFRFC) | 55 |
| Brain Repository | | |
| | McCabe's Complexity | 24 |
| | SQL Complexity | 29 |
| Fat Repository | | |
| | Coupling to Entities (CTE) | 1 |

## 3. The Case Study: The Automated Teller Machine (ATM) Subsystem

This section concerns the development of a Web application named `ATMProject`. The example is carried out by following the steps of the Software Development Process recalled in Section 2.1 and by making use of `xGenerator` (Section 2.2). A similar example is taken into account in [36–38]. The basic *business vocabulary* (i.e., the general concepts) of our example is composed of the following nouns and noun phrases: Customer, Bank, ATM, Bank Account, PIN, Bancomat code, Check Balance, Amount, Deposit Money, Transfer Money, Transaction, Currency, Register ATM, ATM technician, Maintenance and Repair (hardware, firmware or software upgrade).

The *business rules* to be implemented are the following: (a) a customer can hold many bank accounts, while an account is owned by a single customer. (b) An account may be linked to many cards. (c) A customer must be able to make a cash withdrawal from any account linked to the card entered into the ATM; moreover, a customer must be able to make a deposit to any account linked to the card entered into the ATM. (d) Many transactions can be made on an account. (e) A transaction refers to one currency and takes place at a physical ATM. (f) Many maintenances and repairs may happen on ATMs.

Through an ATM, bank customers can perform financial transactions in a public space without the need for a cashier. An ATM session starts when the customer inserts an ATM card into the machine; then, he is asked to enter the Personal Identification Number (PIN). The latter joins the customer to a specific bank account. Then, he is allowed to perform one or more transactions, choosing from a list of alternatives. To authenticate a customer and perform transactions, the ATM subsystem must interact with the bank's database about accounts. Users' authentication is out of scope of the `ATMProject`.

Figure 6 shows the *Business UC Diagram*. It is composed of a single Business UC (`Automatic Teller Machine`) that communicates with the Business Actors (`Bank` and `ATM Technician`) and the Business Worker `Customer`. As usual in UML, business workers are stereotyped as «Business Worker», while business actors are stereotyped as «Business Actor».

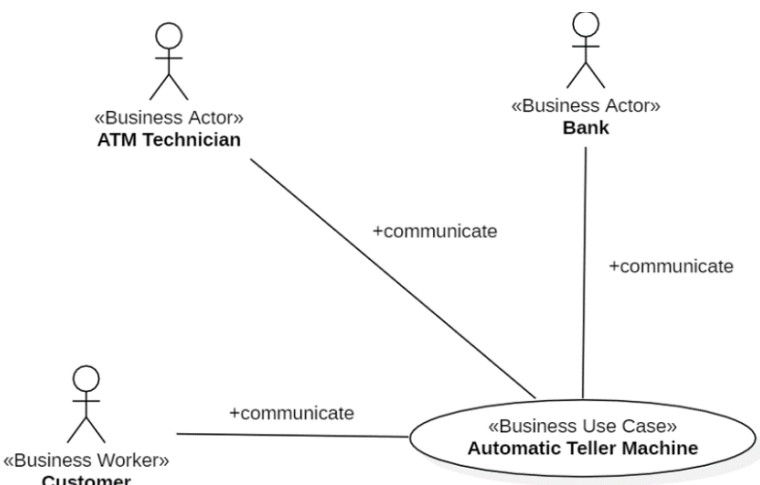

**Figure 6.** The Business UC Diagram.

The customer uses the ATM to Check the Balance of his bank account(s), Deposit Money into a bank account, Withdraw Cash and/or Transfer Money from one account to another. Check Balance, Deposit Money, Withdraw Cash and Transfer Money are "top level" UCs, i.e., Business UC Realizations (Section 2.1). Figure 7 shows the Business UC Realizations that realize the `Automatic Teller Machine` Business UC.

Figures 8–10 show, respectively, the Business UC Realizations the three "roles" of Figure 6 interact with. The Business Objects of the ATM Subsystem are shown in Figure 11. The Business Actors `Bank` and `ATM Technician` (Figure 6) do not give rise to business objects because those actors are managed by the `Login` component of `xGenerator` [14].

The Business Worker `Customer` communicates with the Business UC Realizations `Withdraw`, `Deposit`, `Transfer` and `Check Balance` through the Business Object `Transaction`. The transaction has sign negative for withdrawals and sign positive for deposits. In the case of transfers of money between two accounts, there are two business objects `Transaction` with opposite sign.

The business object `MaintenanceCategory` specifies the six categories of maintenance on the ATMs taken into account in our example, namely: replenishing of cash, ink or printer paper; upgrade of hardware, firmware or software.

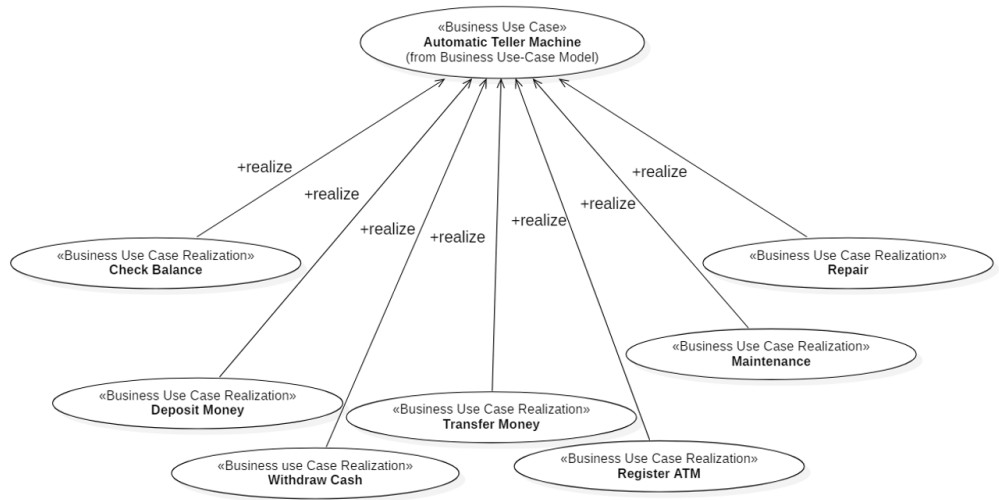

**Figure 7.** The Business UC Realization Diagram.

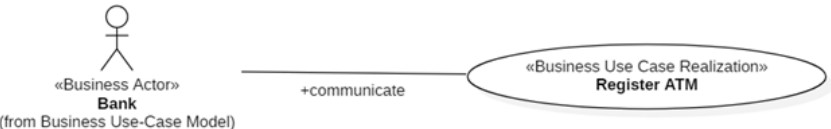

**Figure 8.** The Business UC Realization the Business Actor `Bank` communicates with.

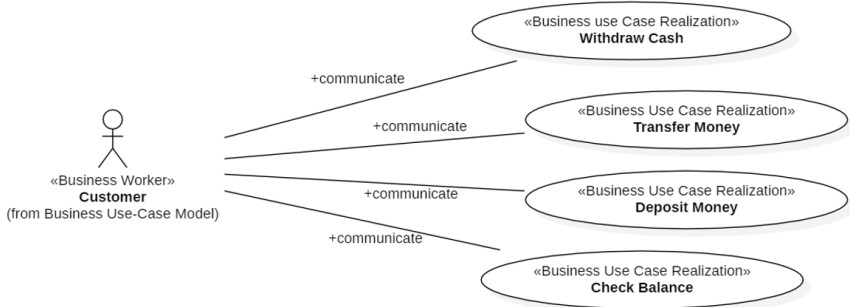

**Figure 9.** The Business UC Realizations the Business Worker `Customer` communicates with.

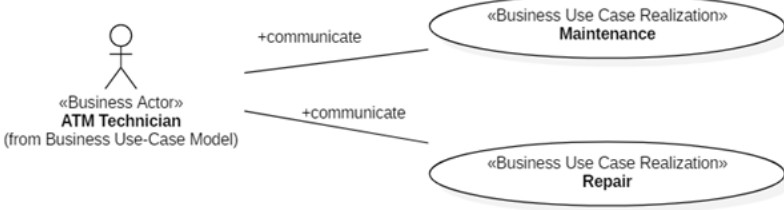

**Figure 10.** The Business UC Realizations the Business Actor `ATM Technician` communicates with.

Figure 12 shows the class diagram that takes into account the Business Objects of Figure 11 and the business rules, while Figure 13 shows the tables of the underling PostgreSQL database.

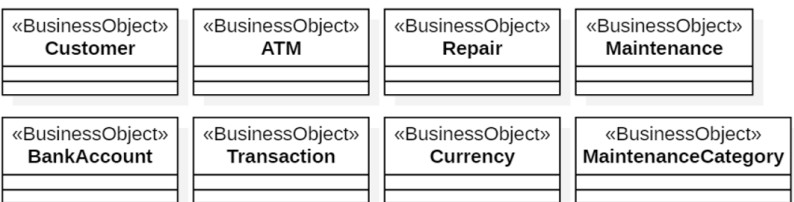

**Figure 11.** The Business object diagram.

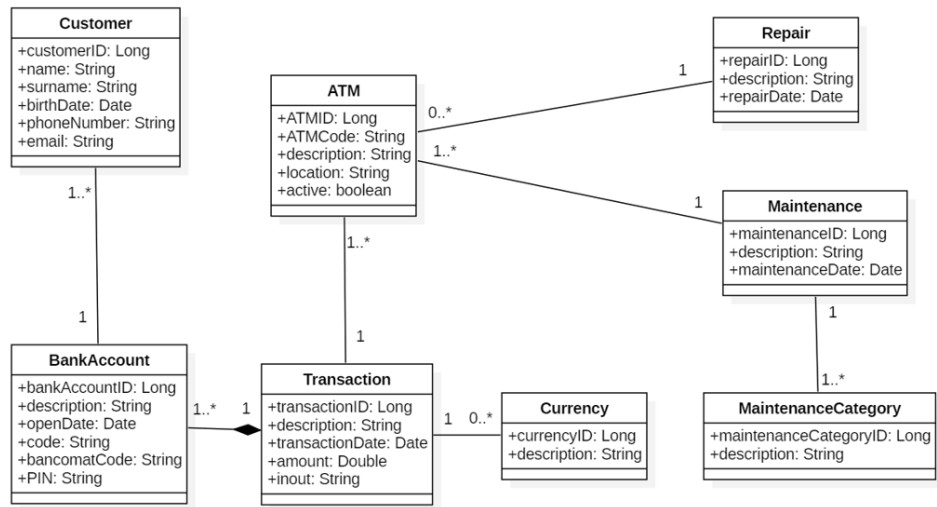

**Figure 12.** The Bean class diagram.

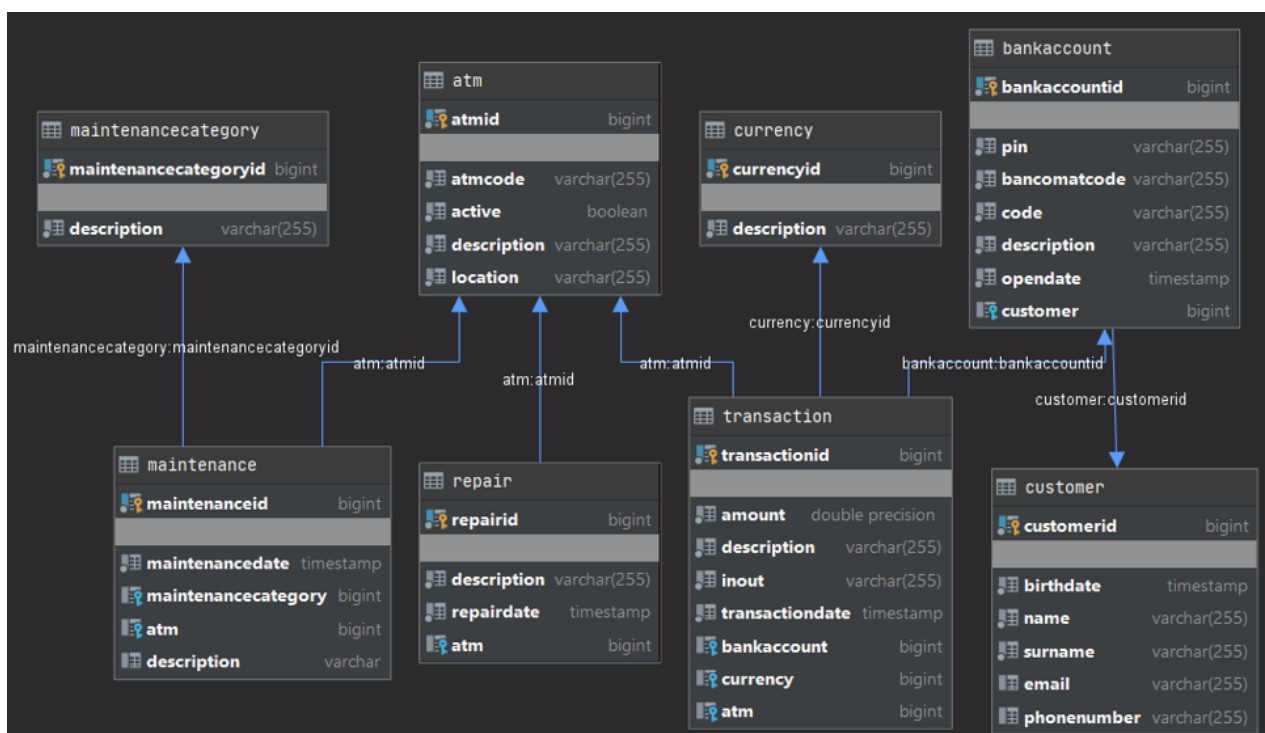

**Figure 13.** The database of the `ATMProject`.

## 4. Method

The quality of the code outputted by `xGenerator` is high "by definition" because it implements a model-driven approach based on MDA and, as it is known, "MDA leverages

models to [. . . ] improve the quality and maintainability of the resulting products" [39] (p. 1). Kulkarni summarizes a long experience of delivering business applications using a model-driven development approach as follows: "In our experience, large application development projects benefited from this approach in terms of technology independence, enhanced productivity and uniformly high code quality." [10] (p. 233). Nonetheless, the objective of our study was the assessment of the quality of the source code of the ATMProject.

ATMProject is composed of 33 classes (7 Controller classes, 8 Bean classes, 4 QueryContainer classes, and 14 View classes) for a total of 2349 lines of Java code. To measure the quality of such a code (with the exception of the View classes), we used the Springlint tool based on Mauricio Aniche's PhD thesis [40]. Springlint compares the classes playing a given role (e.g., Controller) to a benchmark of thousands of classes from 120 Spring MVC systems. Red dark squares mean the class is within the 10% worst classes (classes with highest values) in the benchmark. Light red squares (10–20%), yellow (20–30%), green (30–100%) are the other categories. In addition, the size of the square is proportional to the metric value: the bigger the square the higher the metric value. In practice, we should be worried about red classes. Springlint also detects the six smells of Table 4 in Spring MVC systems.

The MVC pattern implemented by xGenerator (Figure 4) is not identical to Spring MVC (Figure 5). The consequence is that the structure, and hence the content, of the three layers of the code are not identical in the two solutions. Nonetheless, it was possible to use the Springlint tool because the following correspondences exist between the two implementations of the MVC pattern:

- Bean classes match the Spring Entity classes.
- UseCase classes match the Spring Controller classes; but, besides implementing all the actions of the Controller layer of Spring, UseCase classes manage also the navigation of the use cases according to the UML's definition.
- QueryContainer classes match the Spring Repository classes. The methods in both families implement queries against the underlying database; but, while QueryContainer classes may contain multiple QueryObjects referring to different Bean classes, Aniche et al. [16] postulated that Repository classes must contain a single QueryObject linked to a single Bean class.

Coherently with the correspondences listed above, xGenerator tagged the Java classes of the ATMProject with an annotation that specifies their architectural role. The annotations are the following: @Controller, @Repository and @Entity.

## 5. The Results

Figure 14 shows the names of the classes that compose the packages Bean, QueryContainer and Controller of the layers of the ATMProject. The number of the classes in the figure comes from the adopted Software Development Process. The Controller classes are as many as the number of the Business UC Realization (Figure 7). There exists a 1-to-1 correspondence between the Business objects (Figure 11), the database tables (Figure 13), the classes in the class diagram (Figure 12) and the Java Bean classes of the generated code (Figure 14). About the QueryContainer classes of the Model layer (Figure 4), their number and content derive from the project specifications. In the ATMProject, such a number is 4 because it was decided to display information about: (a) the ATMs; (b) the customers and their bank accounts; (c) the ATMs and the undergone repairs; and (d) the ATMs, the undergone repairs and the categories of maintenance they were involved in. Regarding their content, it should be remembered that QueryContainer classes define the filter criteria inside the queries to be executed against the database (Section 2.2). Therefore, in the ATMProject QueryContainerATM accesses 1 table (i.e., ATM); QueryContainerBankAccount accesses 2 tables (i.e., Customer and BankAccount); QueryContainerMaintenance accesses 4 tables (i.e., ATM, Maintenance, MaintenanceCategory and Repair); and QueryContainerRepair accesses 2 tables (i.e., ATM and Repair).

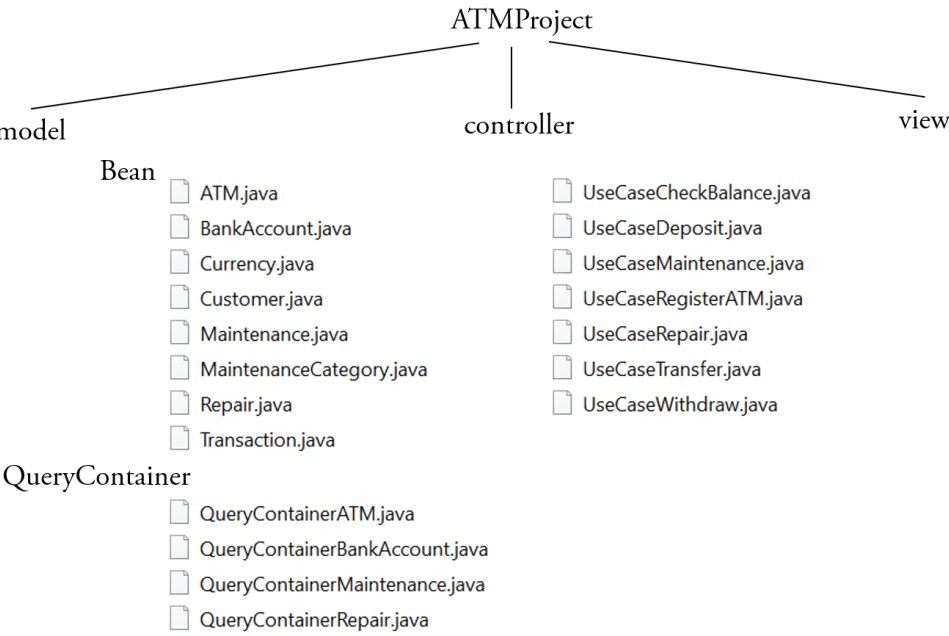

**Figure 14.** The classes of the `ATMProject`.

Tables 6–8 collect the numerical values of the metrics of Table 2 returned by `Springlint` for the classes `Controller`, `Bean` and `QueryContainer`, respectively. The empty boxes in these tables denote the zero value. The output of `Springlint` (Figures 15–17) is an HTML file. We named it `springlint-result ATMProject.html` and made it available at URL www.softwareindustriale.it/ATMProject.html (ATMProject).

**Table 6.** The values of the metrics for the `Controller` classes.

|  | CBO | LCOM | NOM | RFC | WMC |
|---|---|---|---|---|---|
| UseCaseCheckBalance | 7 | 24 | 6 | 27 | 11 |
| UseCaseDeposit | 9 | 24 | 6 | 43 | 10 |
| UseCaseMaintenance | 14 | 6 | 4 | 12 | 16 |
| UseCaseRegisterATM | 12 | 6 | 4 |  | 6 |
| UseCaseRepair | 14 | 6 | 4 | 12 | 16 |
| UseCaseTransfer | 10 | 35 | 8 | 51 | 17 |
| UseCaseWithdraw | 9 | 28 | 7 | 48 | 16 |

**Table 7.** The values of the metrics for the `Bean` classes.

|  | CBO | LCOM | NOM | RFC | WMC |
|---|---|---|---|---|---|
| ATM | 8 | 56 | 12 | 1 | 12 |
| BankAccount | 12 | 120 | 17 |  | 17 |
| Currency | 7 | 11 | 6 | 1 | 6 |
| Customer | 9 | 91 | 15 |  | 15 |
| Maintenance | 12 | 45 | 11 |  | 11 |
| MaintenanceCategory | 7 | 7 | 6 |  | 6 |
| Repair | 11 | 28 | 9 |  | 9 |
| Transaction | 12 | 120 | 17 |  | 17 |

The results are excellent with just one pseudo-exception. In fact, the values of the five metrics for the three different categories of classes are always below the smallest threshold of Table 3. Those values confirm that the classes returned by `xGenerator` are characterised by high cohesion and low coupling (Section 2.2). The pseudo-exception is represented by value LCOM = 35 for the `UseCaseTransfer Controller` class (Figure 15); but, as we can see, value 35 is almost coincident with value 33 of Table 3.

With respect to Table 7, we can say the following. The values of metrics NOM and WMC are identical because the McCabe's cyclomatic complexity equals 1 for each of the methods belonging to the `Bean` classes. The coincidence of values for the classes `BankAccount` and `Transaction` is casual.

With respect to the smells, `Springlint` produced the following answers:

(a)　Are there smells in your Controllers? No!
(b)　Are there smells in your Entities? No!
(c)　Are there smells in your Repositories?

`QueryContainerBankAccount` Fat Repository,
`QueryContainerRepair` Fat Repository,
`QueryContainerMaintenance` Fat Repository.

Aniche et al. [16] (p. 2130) defines Fat Repository as "a Repository managing too many entities" (Table 4); where an Entity represents a domain object (e.g., a bank account) managed by a specific class (in our case a `Bean` class). They called this metric CTE. If CTE > 1 (Table 5), the class is labeled smelly. The interactions of the 4 `QueryContainer` classes in the `ATMProject` with the `Bean` classes are detailed below.

```
QueryContainerATM:
    import model.bean.ATM;
QueryContainerBankAccount:
    import model.bean.BankAccount;
    import model.bean.Customer;
QueryContainerMaintenance:
    import model.bean.ATM;
    import model.bean.Maintenance;
    import model.bean.MaintenanceCategory;
    import model.bean.Repair;
QueryContainerRepair:
    import model.bean.ATM;
    import model.bean.Repair;
```

It follows that, CTE = 1 for the `QueryContainerATM` class, while CTE > 1 for the other three classes. In light of the study of the state-of-the-art, which constitutes the underlying base of the experiments we carried out and the results we got, it follows that to respect the constraint CTE = 1, it is necessary to impose a best practice that implements the refactoring suggestion proposed in [16] when CTE > 1, i.e., move the methods that are related to other Entities to the Repository specific to them. In our case, this means having as many `QueryContainer` classes as the number the database tables. Consequently, complex queries have to be created by assembling elementary queries, each of which has to access only one table. Obviously we can do this but it does not necessarily represent a benefit and in any case it is not a universally shared solution. For example, in [16], the authors carried out an experiment on 100 Spring MVC projects. It is interesting to note that the most common smell in terms of percentage of affected classes was the Fat Repository (20.5%, page 2137). This proves that even Spring MVC programmers often violate the threshold CTE = 1 proposed in [16] because such a constraint is considered by programmers too rigid.

**Table 8.** The values of the metrics for the `QueryContainer` classes.

|  | CBO | LCOM | NOM | RFC | WMC |
|---|---|---|---|---|---|
| QueryContainerATM | 7 | 3 | 3 | 6 | 3 |
| QueryContainerBankAccount | 6 |  | 4 | 6 | 5 |
| QueryContainerMaintenance | 13 | 3 | 3 | 5 | 3 |
| QueryContainerRepair | 12 | 3 | 3 | 6 | 3 |

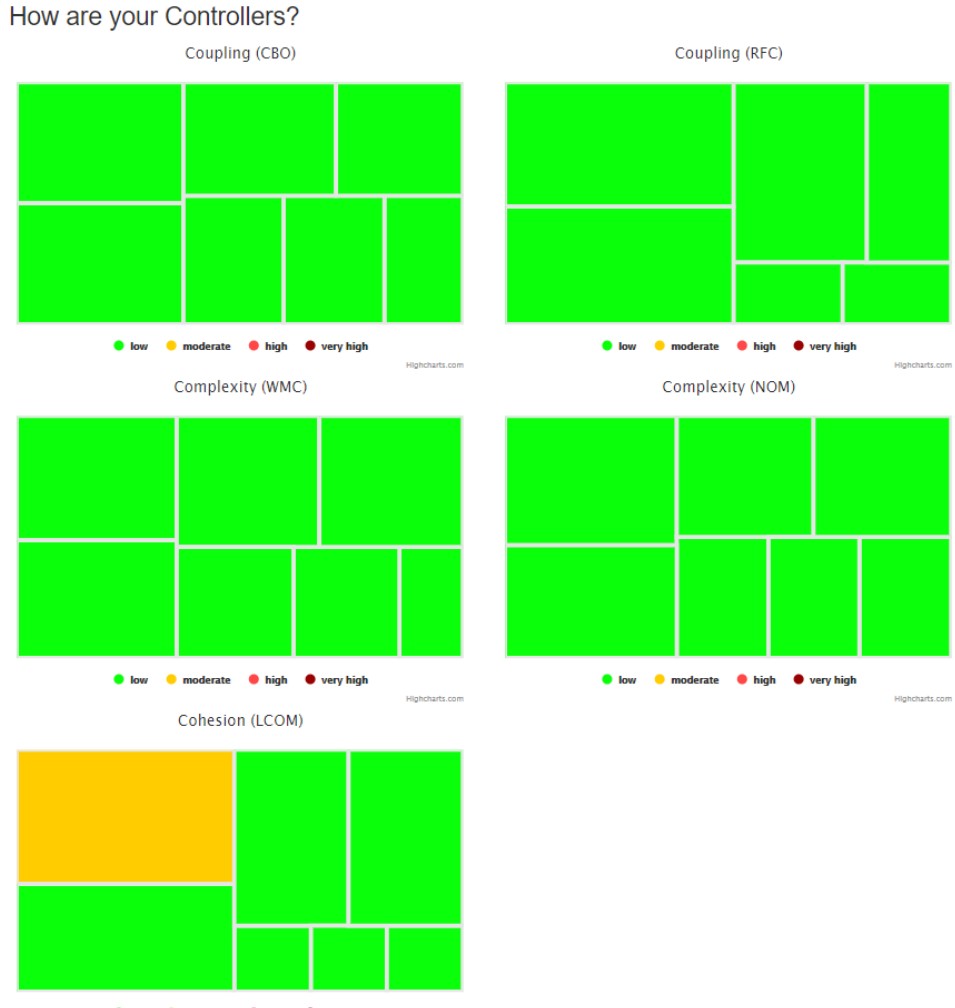

**Figure 15.** The graphical output of `Springlint` with the metrics of Table 2 for the `Controller` classes.

Summarizing, from the experiments carried out, the following can be said. Through `Springlint`, it was possible to investigate the quality of the `Controller` and `Model` classes of `ATMProject`. These are 19 Java classes for a total of 1678 lines of code. The measurements showed that for all classes the values of the metrics proposed in [15] are below the minimum threshold (i.e., the $v_1$ value in Table 3). The measurements also showed that the `Controller` and `Model` classes of `ATMProject` satisfy the thresholds of the six MVC smells part of the catalog in [16], with the only exception of 3 classes out of the 4 being part of the `QueryContainer` sub-layer for which CTE > 1.

*5.1. Threats to Validity*

There is a number of potential threats to validity that could influence the results of our study.

- *Construct validity* is concerned with the relationship between theory and findings. What was measured in our experiments comes from the small set of metrics introduced by Aniche et al. [15]. Those metrics cover many aspects of object-oriented programming; nonetheless, in the future, we will carry out further experiments using a larger set of metrics. Another threat to the construct validity arises because we considered only the six types of code-smells proposed in [16]. As future work, other types of code-smells detection methods will be evaluated.
- *Internal validity* concerns external factors that may impact the experiments' outcome. At the moment, we are not able to exclude that projects much more complicated than

ATMProject may determine results about metrics and smells somehow dissimilar from those obtained. To mitigate this issue, more experiments are needed.

- *Conclusion validity* refers to threats that can impact the reliability of our conclusions. The basic threat in this category comes from the adoption of metrics to assess the quality of the ATMProject. As already pointed out in Section 2.3, the assessments produced by metrics-based detection tools are prone to false-positives because this category of methods depends on the metrics thresholds. In our study, false-positives are prevented because Aniche et al. defined the thresholds by taking into account the *context* [15].
- *External validity* refers to the relevance of the results and their generalizability. We conducted the experiments with a Java software project returned by a proprietary tool (xGenerator); consequently, we cannot claim generalizability of our approach to projects adopting a different programming language. At the same time, it is worth notice that the obtained good results emphasize the relevance to the industry of xGenerator.

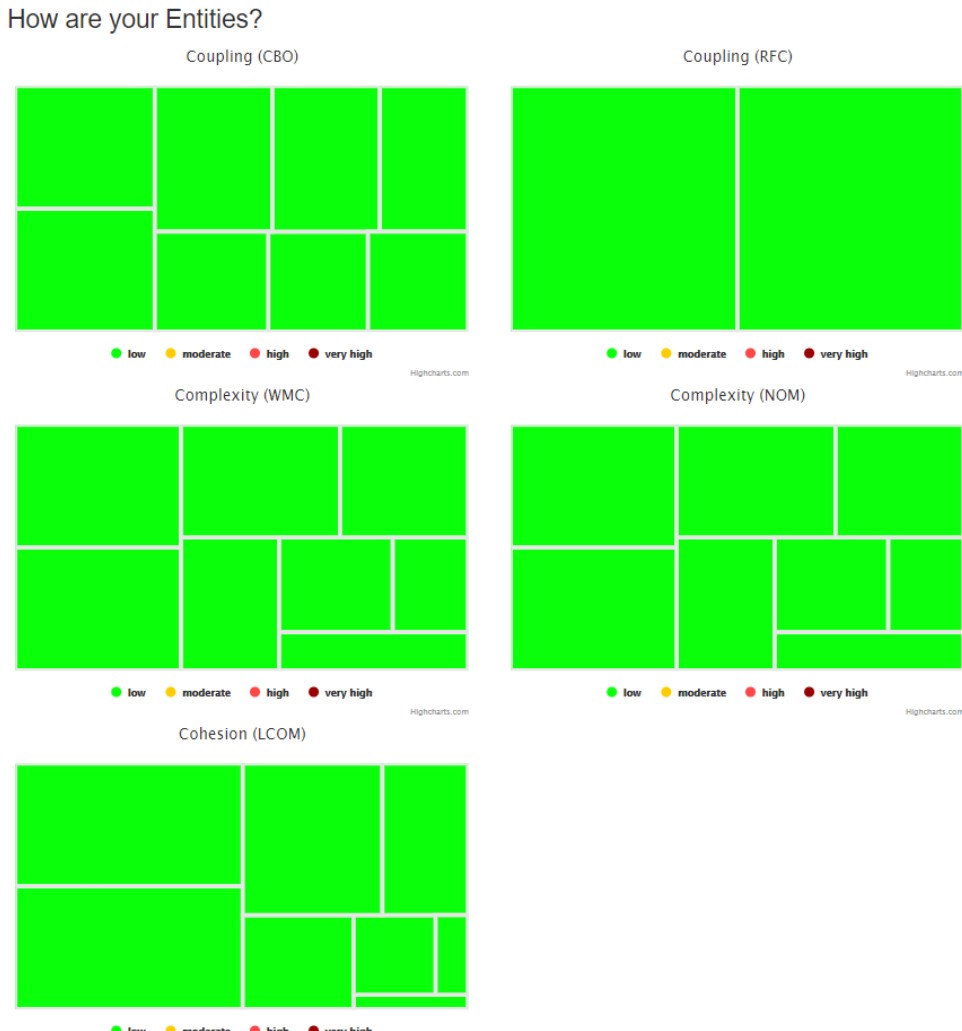

**Figure 16.** The graphical output of Springlint with the metrics of Table 2 for the Entity classes.

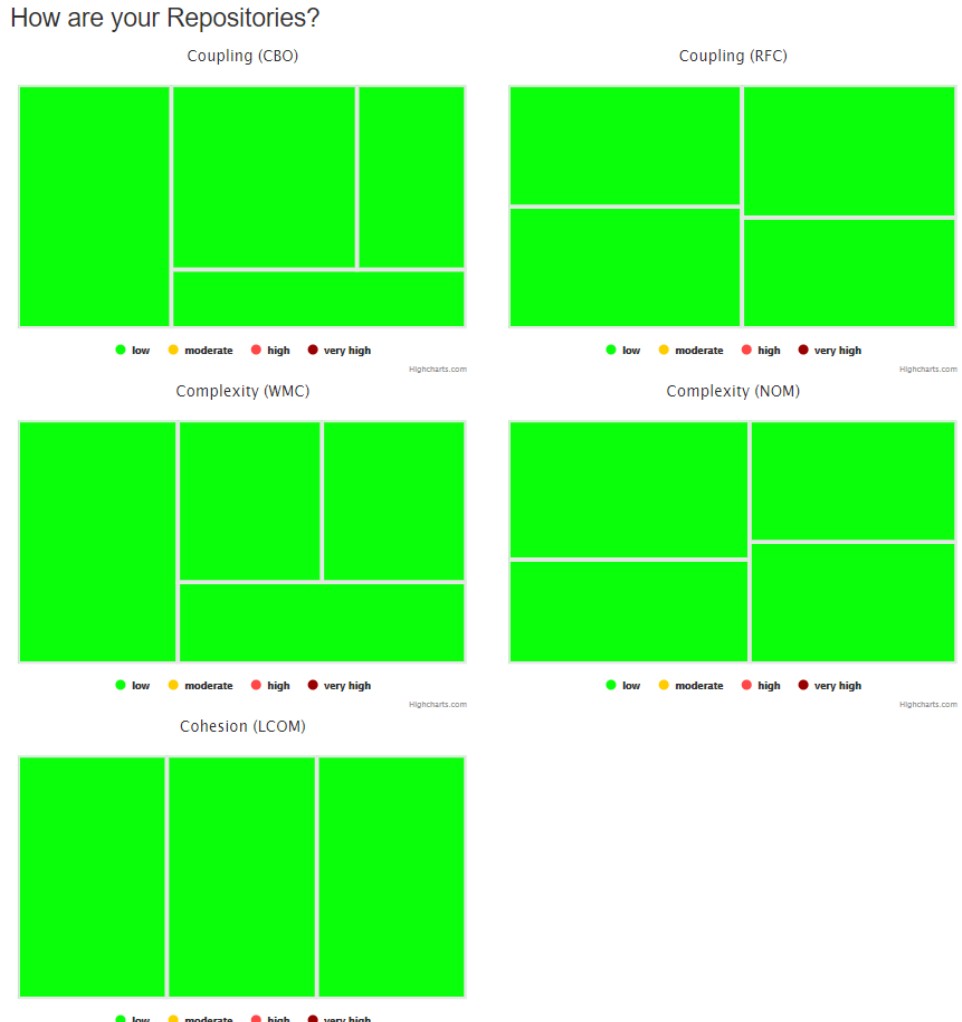

**Figure 17.** The graphical output of `Springlint` with the metrics of Table [2] for the `Repository` classes.

### 6. Conclusions

The aim of the present paper was to assess the quality of the Java source code returned by `xGenerator`: a proprietary tool for the development of MVC Web applications. The experiments were carried out on a real-life case study (`ATMProject`: 19 Java classes for a total of 1678 lines of code) by means of a publicly available tool (`Springlint`). Through `Springlint`, it was possible to investigate the quality of the `Controller` and `Model` classes of `ATMProject`.

First of all, the values of five different code metrics for the `ATMProject` were measured. The measures concerned coupling, cohesion and complexity of the object-oriented code. The selected metrics are able to distinguish the architectural role of the different layers of the MVC pattern. This feature is fundamental since `Controller` classes are quite different from `Model` classes. Accordingly, the threshold of the five metrics changes with regard to the class's architectural role. The positive side effect is the following: in our study, the false-positives (an intrinsic problem of metrics-based detection tools) were absent. The numerical values of the five metrics were always below the threshold. This confirms that the classes returned by `xGenerator` are characterized by high cohesion and low coupling.

The second round of experiments concerned the measures of the value of six MVC smells. The results were good for 16 classes out of 19. The problem with the remaining 3 classes arises because they implement queries against several tables of the underlying database. A practice not well seen by the catalog of code smells proposed by [16]. In the future, we want to read more on this topic since there is still an open debate in the literature.

Overall, the the empirical assessment proved that the code quality is satisfactory. Hence, xGenerator becomes a candidate for being adopted in the IT industry where, besides quality, the use of this tool helps achieving the project goals on time. In the situations where being the first to market is vital to get customers, it is fundamental to release the software within a strict deadline.

**Author Contributions:** Conceptualization, P.D.F.; methodology, G.P.; case study, M.M.; software, R.P.; validation, M.M. and R.P.; formal analysis, P.D.F.; writing, P.D.F.; funding acquisition, G.P. All authors have read and agreed to the published version of the manuscript.

**Funding:** This research was funded by *Software Industriale*.

**Institutional Review Board Statement:** Not applicable.

**Informed Consent Statement:** Not applicable.

**Data Availability Statement:** The code of the ATMProject and the results of the experiments are available at https://www.softwareindustriale.it/ATMProject.html (accessed on 27 January 2021).

**Conflicts of Interest:** The authors declare no conflict of interest.

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
