# Peer review of "Empirical Assessment of the Quality of MVC Web Applications Returned by xGenerator"

_computers, doi:10.3390/computers10020020_

Round 1

Reviewer 1 Report

The abstract should be rephrased and structured according to the guidelines. Referencing authors in the abstract should be omitted.

Introduction: Several abbreviations not introduced in the text.

Line 44: The aim of the paper should be specifically stated in the Introduction section.

Figure 1 should be more informative and enhanced by adding the information about the upper and lower level of generations (where generated code is present and where hand-written code is present), and the differences.

Figure 2: Data base -> DataBase?

Line 282 - 306: Are detailed descriptions significant for this presentation? These descriptions are somehow logical based on the UC names.

Line 374: Can the answers be presented as a figure from the Spinglint?

The Conclusion section should include a short overview of the performed actions, to get the attention of the "sneak preview" reader.

Author Response

General comments:
(a) Introduction and (b) Conclusions “Can be improved”.
The detailed comments are listed below.
1.1
The abstract should be rephrased and structured according to the guidelines. Referencing authors in
the abstract should be omitted.
<<The new abstract is structured according to the guidelines available at MDPI: it has no headings and contains less than
200 words. Moreover, the reference to our previous work has been cancelled.>>
1.2
Line 44:
The aim of the paper should be specifically stated in the Introduction section.
<<See lines 38..46>>
1.3
Figure 1 should be more informative and enhanced by adding the information about the upper and
lower level of generations (where generated code is present and where hand-written code is present),
and the differences.
<<The figure about the Generation gap has been removed.>>
1.4
Figure 2: Data base -> DataBase?
<<Corrected. See (new) Fig.1>>
1.5
Line 282 - 306:
Are detailed descriptions significant for this presentation? These descriptions are somehow logical
based on the UC names.
<<True! Lines mentioned above were removed from the re-submitted manuscript.>>
1.6
Line 374:
Can the answers be presented as a figure from the Spinglint?
<<Done. See (new) Figures 15, 17 and 17.>>
1.7
The Conclusion section should include a short overview of the performed actions, to get the
attention of the "sneak preview" reader.
<<The section has been rewritten entirely as suggested by the reviewer.>>

Reviewer 2 Report

The paper proposes a contribution for the use of Model Driven Architecture (MDA) and Model Driven Engineering (MDE) claiming that in industry their use is still marginal. The paper reports on the evaluation of the quality of the code produced by xGenerator, this can be potentially interesting.

Nevertheless, MDA and MDE are not the only model driven methods, MBSE, MDSEA and for instance recent MDISE can be studied to generalize and better situate used methods, why and this in more general engineering context so not only for code generation.

Moreover, the list of methods for evaluating the quality of object-oriented source code is not complete and exhaustive enough.

The measures proposed in section 3 in the study with metrics and thresholds for MVC web applications are interesting but not sufficiently introduced and explained in section 3.2. It misses a pedagogical way from my point of view. As well, the second case study presents a catalog of six smells suitable for MVC web applications.

The selected metrics were used to assess the quality of code in a real project. The results of the empirical evaluation do not prove that the quality of the code is verified and validated with sufficient explanation and justification.

Et the end, the Java xGenerator platform for the development of enterprise Web applications based on a software development process is still to be better presented to convince readers of the industrial value.

Author Response

 Dear Reviewer,

Reviewer 3 Report

The authors address quality check of the code generated by the application called xGenerator. The work presented here is based on the authors already published work in [15]: G. Paolone, M. Marinelli, R. Paesani, and P. Di Felice: "Automatic Code Generation of MVC Web Applications," Computers, vol. 9, no.3, 56 (2020); https://doi.org/10.3390/computers9030056. There are many overlapping parts/figures in the two articles, which are not addressed properly. It is hard to see, which are the novel parts in the current article.

My comments/remarks to the authors:

  • Please highlight/cite your previous work properly. There are many overlapping in figures (e.g. Fig. 1, 2, etc.).
  • When you refer to [16] and [17] as a basis for your studies, you should also highlight if you agree, support, or disagree with those. Without such remarks, I think we could rename the paper to application only, as there is no assessment.
  • While "code smells" provide you quantitative answers, how do you interpret those? I am missing a discussion on this.
  • From Section 5, the novel part starts, previous parts should be reduced in size as those are mainly survey/repetition.

Author Response

General comments:
(a) Method, (b) Results and (c) Conclusions “Can be improved”.
The detailed comments are listed below.
2.1
The authors address quality check of the code generated by the application called xGenerator. The
work presented here is based on the authors already published work in [15]: Computers, vol. 9,
no.3, 56 (2020).
There are many overlapping parts/figures in the two articles, which are not addressed properly. It is
hard to see, which are the novel parts in the current article.
<<The authors’ idea writing the original manuscript was that of providing an overview of the Software Development Process
described in our previous paper (https://doi.org/10.3390/computers9030056). The consequence was the repetition of
several sentences (about the Software Development Process) from that work, as properly remarked by the reviewer and
the following comment from the Managing Editor:
“[..], we found the content in your manuscript has 25% with your last work in Computers.
[…] Please revise the similarity.”
In hindsight we can say that our idea was not so good since the understanding of the empirical results reported in the
present paper is independent of how the code is generated.
The old Section 2 (Background) and Section 3 (The proposed Software Development Process) have been merged in the
Section 2 of the re-submitted manuscript. In it, the arguments are proposed in a different order and with less details.>>
2.2
When you refer to [16] and [17] as a basis for your studies, you should also highlight if you agree,
support, or disagree with those. Without such remarks, I think we could rename the paper to
application only, as there is no assessment.
<<The reply to this remarks is articulated in three points:
(1) the two important papers ([15] and [16]) by Aniche and his colleagues are the basis of our study. That implicitly
means that we fully agree with them.
(2) The present study concludes the previous research published in MDPI-Computers, vol. 9, no.3, 56 (2020) by adding
the part missing in that paper, namely the assessment of the quality of the code produced by xGenerator. It is true
that the present manuscript doesn’t add new knowledge to the state of art, nonetheless it is important since it
enhances the value of the previous paper. In essence, this second paper should be considered integral part of the
previous one.
(3) We think that it is not fair to label the present paper as an application. It is an empirical study that was possible to
carry out after a careful reading of many pertinent studies. (40 references are listed in the manuscript.)>>
2.3
While "code smells" provide you quantitative answers, how do you interpret those? I am missing a
discussion on this.
<<Lines 260..273 and Lines 310..321 are about the discussion suggested by the reviewer.>>
2.4
From Section 5, the novel part starts, previous parts should be reduced in size as those are mainly
survey/repetition.
<<As explained above, the issue has been solved.>>

Round 2

Reviewer 2 Report

The paper proposes a contribution for the use of Model Driven Architecture (MDA) and Model Driven Engineering (MDE) claiming that in industry their use is still marginal. The paper reports on the evaluation of the quality of the code produced by xGenerator, this can be potentially interesting.

Nevertheless, MDA and MDE are not the only model driven methods, MBSE, MDSEA and for instance recent MDISE can be studied to generalize and better situate used methods, why and this in more general engineering context so not only for code generation.

Moreover, the list of methods for evaluating the quality of object-oriented source code is not complete and exhaustive enough.

The measures proposed in section 3 in the study with metrics and thresholds for MVC web applications are interesting but not sufficiently introduced and explained in section 3.2. It misses a pedagogical way from my point of view. As well, the second case study presents a catalog of six smells suitable for MVC web applications.

The selected metrics were used to assess the quality of code in a real project. The results of the empirical evaluation do not prove that the quality of the code is verified and validated with sufficient explanation and justification.

Et the end, the Java xGenerator platform for the development of enterprise Web applications based on a software development process is still to be better presented to convince readers of the industrial value.

Reviewer 3 Report

The authors have replied to all answers from the reviewers. The quality and presentation of the article has been increased.

Author Response

thanks.

Round 3

Reviewer 2 Report

Reviewer 2 (The late revision)
The paper proposes a contribution for the use of Model Driven Architecture (MDA) and Model Driven Engineering (MDE) claiming that in industry their use is still marginal.
<<The claim is not from the authors but it comes from [13]. Many other scholars stated the same claim.>>
OK
The paper reports on the evaluation of the quality of the code produced by xGenerator, this can be potentially interesting. Nevertheless, MDA and MDE are not the only model driven methods, MBSE, MDSEA and for instance recent MDISE can be studied to generalize and better situate used methods, why and this in more general engineering context so not only for code generation.
Reviewer Round 2: This question is not answered : Nevertheless, MDA and MDE are not the only model driven methods, MBSE, MDSEA and for instance recent MDISE can be studied to generalize and better situate used methods, why and this in more general engineering context so not only for code generation.

Detailed comments
1.
Moreover, the list of methods for evaluating the quality of object-oriented source code is not complete and exhaustive enough.
<<The paper is not a survey about methods for evaluating the quality of object-oriented source code. It reports about an empirical study strictly connected to a previous work of the same authors. So it is not clear why the reviewer ask that we provide a “complete” list of papers about the evaluation of the quality of object-oriented code. The latter is a general topic treated in an endless list of contributions.
Reviewer Round 2: I say not complete and exhaustive enough. So I ask to extend not to have all of them.

In the (new) Section 2.3 (Code Metrics and Code Smells) the paper converges on two very recent proposals that take into account “context” based metrics. In our work context means taking into account the “architectural role” of the classes of the different classes of the three layers of the MVC pattern. Most of previous papers about metrics ignore the context, that is another basic reason why we ignore them.>>
Reviewer Round 2: ok

2.
The measures proposed in section 3 in the study with metrics and thresholds for MVC web applications are interesting but not sufficiently introduced and explained in section 3.2. It misses a pedagogical way from my point of view.
<<Section 3 (in the original manuscript) is about The proposed Software Development Process. So, it is not clear the reviewer’s is remark. The explanations the reviewer (probably) refers to are part of the two papers by (Aniche et al.) we mention in the manuscript. Why should we overlap with them? Plagiarism is forbidden.>>
Reviewer Round 2: I agree that my question is more about recalls from previous works. I ask to better explain measures proposed with metrics and thresholds for MVC web applications. You can present it with you own word. In you abstract you say “The former study is about metrics and thresholds for Web MVC applications,” so it seems important starting point in your paper to show that your approach is different but your background section about that is only 6 lines, to recap your background section 2. is generally too short. Or in case remove the sentence from abstract.
By the way, your comment about plagiarism is not an appropriate author response, I recommend not to use that kind of sarcastic comment that exposes you to being banned from some journals by an editorial team and reviewers with less kindness…

3.
The second case study presents a catalog of six smells suitable for MVC web applications. The selected metrics were used to assess the quality of code in a real project. The results of the empirical evaluation do not prove that the quality of the code is verified and validated with sufficient explanation and justification.
<<The reviewer is right 100%. Section 5 in the revised manuscript deals with this issue.>>
Reviewer Round 2: Ok

4.
Et the end, the Java xGenerator platform for the development of enterprise Web applications based on a software development process is still to be better presented to convince readers of the industrial value.
<<The comment of the reviewer is appropriate. Accordingly, the sentence:
“The results of the empirical assessment prove that the code quality is satisfactory. In turn, this prove the industrial usefulness of xGenerator”
that in original manuscript was present in the Abstract and in the Conclusions has been cancelled.
In the last paragraph of the (new) Conclusions, we say:
“Overall, the empirical assessment proved that the code quality is satisfactory. Hence xGenerator becomes a candidatefor being
adopted in the IT industry […]”. >>
Reviewer Round 2: Ok

Reviewer Round 2: Your figures 15- 17 are all green and very difficult to read!

Author Response

Reviewer Round 2:

1.

MDA and MDE are not the only model driven methods, MBSE, MDSEA and for instance recent MDISE can be studied to generalize and better situate used methods, why and this in more general engineering context so not only for code generation.

<<We thank the Reviewer for this useful hint. It will be taken into account in our future studies, but it remains outside the present one given its scope. We hope he might understand us.>>

2.
The measures proposed in section 3 in the study with metrics and thresholds for MVC web applications are interesting but not sufficiently introduced and explained in section 3.2. It misses a pedagogical way from my point of view.

Reviewer Round 2:

I agree that my question is more about recalls from previous works. I ASK to better explain measures proposed with metrics and thresholds for MVC web applications. You can present it with you own word. In you abstract you say:

“The former study is about metrics and thresholds for Web MVC applications,”

so it seems important starting point in your paper to show that your approach is different.

<<As it is written in the (new) Abstract (Lines 4-6), our manuscript reports the measures concerning a Java code (the ATMProject) generated by following the Software Development Process described in [14] and making use of the companion proprietary tool xGenerator. So, the study does not introduce a new approach for the evaluation of the quality of Web MVC applications.>>

but your background section about that is only 6 lines, to recap your background section 2. is generally too short. Or in case remove the sentence from abstract.

<<The Abstract has been corrected as follows:

(a) the sentence

After a careful literature review about methods for the assessment of the quality of object-oriented source code

has been cancelled because, as correctly pointed out by the reviewer, it is simply not true.

(b) The sentence at Lines 8-10

“A big merit of both these proposals is that they fix the metric thresholds by taking into account the MVC software architecture.”

has been added to explain why we focus just on the two papers from Aniche et al.>>

your comment about plagiarism is not an appropriate author response, I recommend not to use that kind of sarcastic comment that exposes you to being banned from some journals by an editorial team and reviewers with less kindness…

<<I apologize for this sentence. English is not my mother language, for sure this is the main cause of the inappropriate expression. Please forgive me. From the kindness of the writing I can understand the nice person Reviewer 2 is.>>

3.

Reviewer Round 2:

Your figures 15-17 are all green and very difficult to read!

<<True 100%. In the original manuscript these figures were not present, but one of the reviewers requested to show them. The color comes from Springlint and has a semantic meaning (See Line 240). So, we can’t change it.>>

Round 4

Reviewer 2 Report

C1 I insist to consider my comment: "MDA and MDE are not the only model driven methods, MBSE, MDSEA and for instance recent MDISE can be studied to generalize and better situate used methods, why and this in more general engineering context so not only for code generation."

Since you talk about MDE and MDA please recall better MDE, MBSE, MDSEA and for instance recent MDISE

C2 Your figures 15-17 are all green and very difficult to read!

<<True 100%. In the original manuscript these figures were not present, but one of the reviewers requested to show them. The color comes from Springlint and has a semantic meaning (See Line 240). So, we can’t change it.>>

Maybe use the contrast to better reveal the lines between the sqaures.

Author Response

C1

I insist to consider my comment:

"MDA and MDE are not the only model driven methods, MBSE, MDSEA and for instance recent MDISE can be studied to generalize and better situate used methods, why and this in more general engineering context so not only for code generation. Since you talk about MDE and MDA please recall better MDE, MBSE, MDSEA and for instance recent MDISE.”

We reject Comment 1 for the three reasons detailed below. We really hope that the Reviewer can understand our reasons. Of course, we thanks him for the kind hint.

1) Our work is experimental, not methodological. This is the case, for instance, of Ref. [14] (our previous work that this paper completes). In [14], we wrote:

“Model-Driven Development (MDD), Model-Driven Engineering (MDE), Model-Driven Software Development (MDSD),

and MD* are terms used for referring to the existing approaches to model-driven development.”

2) Comment 1 from the Reviewer is ignored in studies already published. Three relevant examples are given below, but there are many others.

Ex.1

Mirco Franzago, Davide Di Ruscio, Ivano Malavolta, and Henry Muccini:

Collaborative Model-Driven Software Engineering: a Classification Framework and a Research Map.

IEEE Transactions on Software Engineering, vol. 44, no. 12, dec. 2018.

In such a study, authors use only the term MDSE (Model-Driven Software Engineering) to refer to the wide domain of Software Engineering methods that use models as first-class entities for describing specific aspects of a software system. MBSE, MDSEA and MDISE are not mentioned.

MDE in our paper corresponds to MDSE in [Franzago et al., 2018].

Ex.2

  1. Ciccozzi, I. Malavolta, and B. Selic:

Execution of UML models: a systematic review of research and practice.

Software & Systems Modeling, vol. 18, pp. 2313–2360 (2019), https://doi.org/10.1007/s10270-018-0675-4.

In Ref. [2], authors use only the acronym MDE to refer to the existing software developments proposals at the core of which is the notion that models serve as primary artifacts in development.

We did the same.

Ex.3 (the Ref. [5])

  1. Kahani, M. Bagherzadeh, J.R. Cordy, J. Dingel, and D. Varró:

Survey and classification of model transformation tools.

Software & Systems Modeling, vol. 18, pp. 2361–2397 (2019)

https://doi.org/10.1007/s10270-018-0665-6.

Reference [5] is an up-to-date survey and classification study about 60 model transformation tools published on a relevant journal. In [5], Kahani et al. write that:

“Model-driven engineering (MDE) is a growing field that advocates the use of models during the entire development process of a system.”

Accordingly, they do not mention at all the relevant variations of MDE as, for instance, those cited by Reviewer 2 (namely: MBSE, MDSEA, and MDISE).

Analogously, in our empirical work (where we refer to just one transformation tool: xGenerator) we think that mentioning MDE might be sufficient since such a term acts as an “umbrella term” for any model-driven approaches to software development.

C2

Your figures 15-17 are all green and very difficult to read!

True 100%. In the new figures the boundary of the geometries (squares and rectangles) are ticker. However, the paper provides the link to the original HTML file returned by SpringLint. It offers an optimal quality.
